# The Metrological Traceability, Performance and Precision of European Radon Calibration Facilities

**DOI:** 10.3390/ijerph182212150

**Published:** 2021-11-19

**Authors:** Thomas R. Beck, Andrei Antohe, Francesco Cardellini, Alexandra Cucoş, Eliska Fialova, Claudia Grossi, Kinga Hening, Jens Jensen, Dejan Kastratović, Matej Krivošík, Patrick Lobner, Aurelian Luca, Franz Josef Maringer, Nathalie Michielsen, Petr P. S. Otahal, Luis Quindós, Daniel Rábago, Carlos Sainz, László Szűcs, Constantin Teodorescu, Cathrin Tolinsson, Cornel Liviu Tugulan, Tuukka Turtiainen, Arturo Vargas, Josef Vosahlik, Goran Vukoslavovic, Hannah Wiedner, Katarzyna Wołoszczuk

**Affiliations:** 1Federal Office for Radiation Protection, 10318 Berlin, Germany; 2Institutul National de Cercetare-Dezvoltare Pentru Fizica si Inginerie Nucleara “Horia Hulubei”, 077125 Ilfov County, Romania; antohe@nipne.ro (A.A.); aluca@nipne.ro (A.L.); constantin.teodorescu@nipne.ro (C.T.); liviu.tugulan@nipne.ro (C.L.T.); 3ENEA-INMRI, 00123 Roma, Italy; francesco.cardellini@enea.it; 4“CONSTANTIN COSMA” Radon Laboratory, Faculty of Environmental Science and Engineering, Babes—Bolyai University, 400294 Cluj-Napoca, Romania; dinualexandra2007@gmail.com (A.C.); szacsvaikinga@gmail.com (K.H.); 5Státní Ústav Jaderné, Chemické a Biologické Ochrany, 262 31 Milin, Czech Republic; fialovaeliska@sujchbo.cz (E.F.); otahal@sujchbo.cz (P.P.S.O.); vosahlik@sujchbo.cz (J.V.); 6Laboratory of 222Rn Studies, Institut de Tècniques Energètiques, Universitat Politècnica de Catalunya, 08028 Barcelona, Spain; claudia.grossi@upc.edu (C.G.); arturo.vargas@upc.edu (A.V.); 7SwedishRadiation Safety Authority, 171 16 Stockholm, Sweden; jens.jensen@ssm.se (J.J.); cathrin.tolinsson@ssm.se (C.T.); 8Bureau of Metrology, Podgorica 81000, Montenegro; dejan.kastratovic@metrologija.gov.me (D.K.); goran.vukoslavovic@metrologija.gov.me (G.V.); 9Slovak Institute of Metrology, Department of Ionizing Radiation, 842 55 Bratislava, Slovakia; krivosik@smu.gov.sk; 10Physikalisch-Technischer Prüfdienst, Bundesamt für Eich- und Vermessungswesen, 1160 Vienna, Austria; patrick.lobner@bev.gv.at (P.L.); franz.josef.maringer@tuwien.ac.at (F.J.M.); hannah.wiedner@wien.gv.at (H.W.); 11Institut de Radioprotection et de Sûreté Nucléaire, 92262 Paris, France; nathalie.michielsen@irsn.fr; 12Radon Group, Laboratory of Environmental Radioactivity of the University of Cantabria, 39011 Santander, Spain; quindosl@unican.es (L.Q.); daniel.rabago@unican.es (D.R.); sainzc@unican.es (C.S.); 13Budapest Főváros Kormányhivatala, 1024 Budapest, Hungary; szucs.laszlo@bfkh.gov.hu; 14Radiation and Nuclear Safety Authority, 00880 Helsinki, Finland; tuukka.turtiainen@stuk.fi; 15Central Laboratory for Radiological Protection, 03-194 Warsaw, Poland; woloszczuk@clor.waw.pl

**Keywords:** radon, interlaboratory comparison, radon activity concentration, calibration, metrological traceability

## Abstract

An interlaboratory comparison for European radon calibration facilities was conducted to evaluate the establishment of a harmonized quality level for the activity concentration of radon in air and to demonstrate the performance of the facilities when calibrating measurement instruments for radon. Fifteen calibration facilities from 13 different European countries participated. They represented different levels in the metrological hierarchy: national metrology institutes and designated institutes, national authorities for radiation protection and participants from universities. The interlaboratory comparison was conducted by the German Federal Office for Radiation Protection (BfS) and took place from 2018 to 2020. Participants were requested to measure radon in atmospheres of their own facilities according to their own procedures and requirements for metrological traceability. A measurement device with suitable properties was used to determine the comparison values. The results of the comparison showed that the radon activity concentrations that were determined by European calibration facilities complying with metrological traceability requirements were consistent with each other and had common mean values. The deviations from these values were normally distributed. The range of variation of the common mean value was a measure of the degree of agreement between the participants. For exposures above 1000 Bq/m^3^, the variation was about 4% for a level of confidence of approximately 95% (k=2). For lower exposure levels, the variation increased to about 6%.

## 1. Introduction

The European Council Directive 2013/59/EURATOM requires that the EU member states introduce regulations for protection against radon exposure in homes and at workplaces [1]. In this context, the European metrology institutes are required to create a harmonized quality level for radon activity concentrations. The realization of the measurement quantity with a high degree of agreement between calibration bodies ensures that measurements are comparable and results are mutually recognized in the EU member states. The term *radon* used in the EU legal act, as well as in this work, refers to the radionuclide ^222^Rn.

In the framework of the European Metrology Programme for Innovation and Research (EMPIR), the project *Metrology for Radon Monitoring (MetroRADON)* was initiated, which included an interlaboratory comparison to evaluate the metrological traceability of European radon calibration facilities and to demonstrate their performance and precision when calibrating measurement instruments for radon in the range from 300 to 10,000 Bq/m^3^.

Calibration services from the different EU member states, which preferably represent the respective national reference for the quantity of radon activity concentration in air, were encouraged to participate in the comparison.

The interlaboratory comparison was conducted by the German Federal Office for Radiation Protection (BfS, coordinator) and took place from 2018 to 2020. The participants were requested to measure the atmospheres of their own facilities according to their own procedures and requirements for metrological traceability. The comparison value was determined with a measurement device that had appropriate metrological characteristics and was made available to the participants in turn. Differences in the comparison values demonstrated the differences of the participants in the measurement of the radon activity concentration. The measure of precision was considered to be the degree of agreement between participants in the determination of the quantity. The interlaboratory comparison also showed the uncertainties when passing on the quantity to third parties through the calibration of devices.

In total, 15 calibration facilities from 12 different countries of the European Union and one from Montenegro participated in the interlaboratory comparison. Table 1 presents the calibration facilities that were involved in the comparison.

The pool of participants encompassed seven national metrology institutes and designated institutes (BEV-PTP, STUK, BFKH, ENEA, IFIN-HH, MNE, SMI), five national authorities for radiation protection (BfS, SUJCHBO, IRSN, CLOR, SSM) and three participants from universities (UBB, LaRUC, UPC).

The considerable number of participants from various European countries with different positions in the metrological hierarchy and thus different positions in the metrological traceability chain allowed for a representative evaluation of the performance and precision in the calibration of measurement instruments for radon.

## 2. Organization and Methodology

### 2.1. Procedure of Interlaboratory Comparison

The basic design of the interlaboratory comparison was developed in consultation with the members of the advisory group, which consisted of the EMPIR project collaborators. An agreed protocol for the comparison was handed out to each participant in advance. It informed about the procedure of the comparison, as well as the handover and handling of the comparison device.

The comparison device was sent to each participant in turn. It was made available to the participant for a predefined duration in order to perform the exposure measurements. After completing the exposure measurements, the device had to be returned to the coordinator. In addition to the exposure data, participants were asked to report data on the temperature, air humidity and air pressures that prevailed during the exposure measurements.

### 2.2. Comparison Device

The comparison device was used to transfer the comparison value for the measurements at different locations and levels. The device did not embody the comparison reference value.

The coordinator selected an electronic instrument of type AlphaGUARD PQ 2000 PRO TTL. This type of device is a standard instrument for the measurement of radon activity concentrations. The instrument is robust and reliable under various environmental conditions and is easy to use. Measurement results are stored safe from manipulation in its internal memory with sufficient capacity for the comparison exercises. The instrument was operated in the diffusion mode with an integration time of 10 min.

The comparison device was calibrated in the facilities of the coordinator at different radon activity concentrations in the range between 300 and 12,000 Bq/m^3^. Calibrations were performed before, during and after the interlaboratory comparison [2]. Taking the uncertainty into account, a constant calibration factor was obtained over the whole investigated range, which pointed to the linear relationship between the indicated value and the radon activity concentration in air. Similarly, no change in the calibration factor was observed over the comparison period, implying that the measurement characteristics of the instrument were constant, allowing for equal conditions for each participant.

The calibrations were flanked by regular background measurements. For this purpose, the device was enclosed in a volume that was flushed with low-radon air. Low-radon air was obtained from pressurized cylinders in which the air had previously been stored for a longer period. The resulting radon concentration in the volume was considered to be negligible (zero) and the device indicated the datum error for zero value of radon activity concentration. The background of the comparison device was measured before each run. It was constant throughout the comparison period and was determined to be 4 ± 5 Bq/m^3^, which had a negligible effect on the measurement results. The attributed uncertainty was the standard uncertainty. The AlphaGUARD operates with automatic background correction. Due to stochastic measurement effects that are not taken into account in the automatic background correction, the device provides measured values for the background that can also be negative.

Visual inspections of the comparison device for damage, including the diffusion filter, verification of functionality and checking of the set measurement parameters (e.g., calibration factor) supplemented the regular checks before the instrument was used for the next run.

### 2.3. Exposure Levels

Within the specified study range between 300 and 10,000 Bq/m^3^, 3 different exposure levels with low, medium and high radon activity concentrations were defined for the comparison. The nominal values of the radon activity concentrations are given in Table 2.

The value of 1000 Bq/m^3^ was already included in a previous comparison of calibration facilities for radon activity concentrations, which was carried out within the framework of the Euromet Project 657 [3].

In practice, the participants could not exactly adhere to the specified nominal values. Therefore, deviations from the nominal values were accepted within which the respective activity concentrations were expected.

With the exception of a few participants, most of the participants were able to meet these requirements. The main reasons for not achieving the nominal radon activity concentrations within their accepted deviations were generally:The participants were not able to keep the activity concentration constant over the duration of exposure, as the activity concentration decreased over time, mainly due to radioactive decay;The radon sources that were available in the participant’s laboratories and/or the methods used to create the radon atmosphere were not suitable for reaching the predetermined concentrations.

The assessment of the degree of agreement between participants was carried out only for the results that were obtained at exposures that were within the accepted deviations. However, the results of measurements outside the accepted deviations from the nominal values were not excluded from consideration and are referred to in the following as singular exposures. They complemented the conclusions of this study by supporting its extension to the entire range of radon concentrations from low to high levels.

### 2.4. Methods for Processing the Results

The investigated quantity, which made the participant’s performance comparable, was the ratio R of the radon activity concentration $C_{\mathrm{RefLab}}$, which was reported by the participant as the mean value for the relevant exposure period and the mean radon activity concentration $C_{\mathrm{CD}}$, which was measured during the same period with the comparison device:(1)$$R=\frac{C_{\mathrm{RefLab}}}{C_{\mathrm{CD}}}$$

The standard uncertainty ∆R=u(R) was calculated from the propagation of uncertainty from Equation (1). The relative uncertainty is given by
(2)$$\frac{∆R}{R}=\sqrt{{\left(\frac{∆C_{\mathrm{RefLab}}}{C_{\mathrm{RefLab}}}\right)}^{2}+{\left(\frac{∆C_{\mathrm{CD}}}{C_{\mathrm{CD}}}\right)}^{2}}$$

$∆C_{\mathrm{RefLab}}=u\left(C_{\mathrm{RefLab}}\right)$ represents the standard uncertainty as reported by the participant and was determined according to its own procedure. The reported uncertainty included the statistical variation from repeated observations (type A evaluation of standard uncertainty) and contributions from other sources, in particular from data provided in the calibration and other certificates (type B evaluation of standard uncertainty) [4]. However, the procedure that was used by participants to calculate the measurement uncertainties was not evaluated as part of this interlaboratory comparison. $∆C_{\mathrm{CD}}$ is the uncertainty of the mean radon activity concentration, which was determined by the comparison device. Since the comparison device did not embody the comparison reference value, only the type A uncertainty given by the standard deviation of the mean, namely, *s*$\left(C_{\mathrm{CD}}\right)$, was considered:(3)$$∆C_{\mathrm{CD}}=s\left(C_{\mathrm{CD}}\right)=\sqrt{\frac{\sum {\left(C_{\mathrm{CD},j}-C_{\mathrm{CD}}\right)}^{2}}{n\left(n-1\right)}}$$

$C_{\mathrm{CD},j}$ is the *j*th of *n* measurements that were taken with the comparison device. Other contributions to the uncertainty, particularly from calibration factors, were not included. This was due to the essential requirement for the comparison device to provide an indication that depended linearly on the value of the radon activity concentration. The initial investigations of the comparison device showed that the linearity could be assumed over the entire range up to a radon activity concentration of 10,000 Bq/m^3^.

It should be noted that the simple averaging of the measurements performed with the comparison device and the use of Equation (3) was valid if the activity concentration was kept constant during the relevant exposure period. If this could not be ensured by the participant, the change in activity concentration over time must be well known. In such cases, the participant had to provide information on how to determine the mean radon activity concentration $C_{\mathrm{CD}}$ from the readings of the comparison device. In general, the radon activity concentration that was established in a confined atmosphere decreased due to radioactive decay and, thus, the value for $C_{\mathrm{CD}}$ at the reference time $t_{\mathrm{ref}}$ was obtained using
(4)$$C_{\mathrm{CD}}\left(t_{\mathrm{ref}}\right)=\frac{1}{n}\;\underset{j}{\sum }C_{\mathrm{CD},j}\;e^{-\lambda \left(t_{\mathrm{ref}}-t_{j}\right)}$$

Equation (4) represents the average of the measured values corrected for the exposure time. The parameter $t_{j}$ represents the measurement time of $C_{\mathrm{CD},j}$ and *λ* represents the decay constant of radon. Equation (4) must be modified if the rate of decrease differs from that of radioactive decay.

### 2.5. Calculation of the Uncertainty-Weighted Average Ratio

When $R_{i}$ denotes the ratio R calculated for the *i*th of *n* participants and $u_{i}$ is the standard uncertainty attributed to $R_{i}$, the uncertainty-weighted average ratio $R_{w}$ is determined using
(5)$$R_{w}=\frac{\frac{R_{1}}{u_{1}^{2}}+\cdots +\frac{R_{n}}{u_{n}^{2}}}{\frac{1}{u_{1}^{2}}+\cdots +\frac{1}{u_{n}^{2}}}=\overset{n}{\underset{i=1}{\sum }}w_{i}R_{i}$$

The parameter $w_{i}$ represents the weight for ratio $R_{i}$:(6)$$w_{i}=\frac{\frac{1}{u_{i}^{2}}}{\sum_{i=1}^{n}\frac{1}{u_{i}^{2}}}$$

The weights are calculated from the reciprocal squared standard uncertainties of Equation (2). It follows that results with lower uncertainties are weighted higher than results with high uncertainties when determining the average ratio. The variance of $R_{w}$ is calculated as follows [5,6]:(7)$$\sigma^{2}\left(R_{w}\right)=u^{2}\left(R_{w}\right)=\frac{1}{\frac{1}{u_{1}^{2}}+\cdots +\frac{1}{u_{n}^{2}}}=\frac{1}{\sum_{i=1}^{n}\frac{1}{u_{i}^{2}}}$$

## 3. Results

### 3.1. Laboratory Reference Devices and the Compilation of the Results

Most participants used an AlphaGUARD-type device as the laboratory reference instrument for the radon activity concentrations. Two of these participants additionally performed measurements with scintillation chambers. The ATMOS 12DPX was utilized by two participants and a Radon Scout by one participant. AlphaGUARD and ATMOS use ionization chambers (single- or multi-wire) for radiation detection. The Radon Scout deploys high-voltage enhancement and alpha pulse counting using a semiconductor detector. Unlike AlphaGUARD and Radon Scout, which operate in diffusion mode, the ATMOS-type device operates in flow-through mode.

The vast majority of the participants were able to show the metrological traceability of the quantity through an unbroken chain of calibrations at recognized bodies. Two participants traced their measurements back through factory calibration.

Factory calibration is a service from the manufacturer that provides the instrument with an initial calibration before delivery. Although the manufacturers also trace their measurements back to recognized bodies, compliance with quality management standards and metrological requirements need not be demonstrated. The reported results revealed that, in particular, for participants who used factory calibration, the attributed measurement uncertainties were not consistent with the deviation from the collective average. It is shown below that this has consequences for the use of these results in the assessment of the interlaboratory comparison.

Figure 1 shows the ratios $R_{i}$ representing the radon activity concentrations as measured by the participants in relation to the corresponding readings of the comparison device according to Equation (1). The error bars represent the standard uncertainties according to Equation (2). The results from participants who traced back their measurements using factory calibration are not included in Figure 1.

### 3.2. Consistency Check

A check of mutual consistency is required for interlaboratory comparisons using the BIPM consultative committee CCQM [5] to test the hypothesis that the participants have a collective mean value and that the deviations from this value are normally distributed.

The consistency check is performed using a chi-squared test over the number of *n* measurements (or participants). The observed test parameter $\chi_{\mathrm{obs}}^{2}$ is calculated using
(8)$$\chi_{\mathrm{obs}}^{2}=\overset{n}{\underset{i=1}{\sum }}{\left(\frac{R_{i}-R_{w}}{u_{i}}\right)}^{2}$$

According to CCQM [5], the test parameter is compared with the quantile of the chi-squared distribution for the significance level 1−α with α=0.05. The following decisions have to be made:If $\chi_{\mathrm{obs}}^{2}<n-1$, the results are mutually consistent and the uncertainties account fully for the observed dispersion of the values;If $n-1\leq \chi_{\mathrm{obs}}^{2}<\chi_{0.05;n-1}^{2}$, the data provide no strong evidence that the reported uncertainties are inappropriate, but there remains a risk that additional factors are contributing to the dispersion;If $\chi_{\mathrm{obs}}^{2}>\chi_{0.05;n-1}^{2}$, the data should be considered as mutually inconsistent.

The results of the consistency checks are summarized in Table 3. The tests were performed for each exposure level and for the complete data set of all levels including singular exposures. The two participants who traced their measurements back through factory calibration are not included in the results of the consistency check.

Table 3 shows that for each exposure level, the observed test parameter was below the tabulated value $\chi_{0.05;n-1}^{2}$. Therefore, the conclusion can be drawn that the results were mutually consistent. There was no evidence of significant inconsistencies for each of the individual radon levels and the overall exposure range. The uncertainties fully accounted for the observed dispersion of the values.

For the radon level of 400 Bq/m^3^, the test parameter was greater than n−1 at the stated significance level. The higher value of the observed test parameter was caused by results that showed increased deviations from the average ratio without a corresponding uncertainty being assigned to them. In these cases, the uncertainties that were attributed by some participants might have been too small for the observed deviation from the mean value.

The presented consistency of the data set failed when the results of the two participants who traced their measurements back through factory calibration were included in the data set. To ensure the consistency of the data set and to maintain the degree of representativeness of the intercomparison, the data from the two participants were not included in the derivation of the average ratio and, thus, the comparison reference value. The coordinator (from BfS) was also not considered further due to his special position as part of the supervising laboratory.

### 3.3. The Uncertainty-Weighted Average Ratio

Table 4 shows the uncertainty-weighted average ratio $R_{w}$ for the different exposure levels. $R_{w}$ is calculated according to Equation (5). The square root of the variance from Equation (7) is the standard uncertainty $u\left(R_{w}\right)$. The values of the average ratio obtained for the various exposure levels agreed very well, taking into account the standard uncertainties.

Assuming that the comparison device represents the weighted collective average radon activity concentration for each exposure level, the average ratio would be compensated, resulting in $R_{w}=1$. However, the calculated values for $R_{w}$ showed a bias of about 1.5% above the expected compensation value. The bias was caused by the comparison device due to the calibration of the device at the coordinator’s facility and indicated the coordinator’s deviation in the measurement of the quantity from the collective mean. The measurements of the comparison device were, on average, 1.5% lower than the weighted average radon activity concentration that was measured by the participants for the respective exposure level. It was observed that the average ratio $R_{w}$ varied only slightly for the different exposure levels, confirming the performance and stability of the comparative measurements.

## 4. Discussion

### 4.1. The Key Comparison Reference Value and the Dispersion of Measurement Values

The key comparison reference value (KCRV) is the value of the quantity representing the specific property of the material under consideration [5]. The specific property that was under consideration in this interlaboratory comparison was the activity concentration of radon in air. However, the single radon activity concentrations in the atmospheres that were measured at participants’ facilities differed between participants. Moreover, three different main levels of radon activity concentration were measured by each participant, covering a large range over more than one magnitude.

The comparison device that was provided by the coordinator was used as a comparator to normalize the different radon activity concentrations that were established by the participants and thus allowed for comparability of the respective measurements of the quantities. As the comparison device is characterized by an indication, which is verifiably linear over the entire range, the comparison of the different radon activity concentrations found in the participant’s facilities was made possible by their ratio to the indication of the comparison device, as is given by Equation (1). The average ratio $R_{w}$, which can be deemed to be the KCRV, was calculated from the single ratios according to Equation (5). The consequences were as follows:The observed $R_{w}$ had a bias of about 1.5% (Table 4) compared to the expected value of $R_{w}=1$, which would result if the comparison device were to represent the uncertainty-weighted collective mean radon activity concentration and, thus, all individual deviations were compensated.The variance of $R_{w}$ that was calculated using Equation (7) led to the small values for $u\left(R_{w}\right)$ given in Table 4. As was also shown in other studies [6,7,8], the reciprocal square root of the sum of the weights becomes too small with an increasing number of participants such that many laboratories fall outside the uncertainty interval. It is therefore assumed that this parameter is not an appropriate measure of the degree of agreement between the participants.

To overcome the disadvantages in terms of $R_{w}$ and also to eliminate the impact of the comparison device, a modified ratio, namely, $R_{i}^{*}$, for the *i*th participant is formed by
(9)$$R_{i}^{*}=\frac{R_{i}}{R_{w}}$$

The rationale behind this modified ratio, now considered as a new comparison value, is that the expectation value $E\left(R_{i}^{*}\right)$ that is obtained from the weighted sum over each participant is equal to 1:(10)$$E\left(R_{i}^{*}\right)=\overset{n}{\underset{i=1}{\sum }}w_{i}R_{i}^{*}=\frac{1}{R_{w}}\;\overset{n}{\underset{i=1}{\sum }}w_{i}R_{i}=1$$

The weights $w_{i}$ are given by Equation (6). Equation (10) implies that the values $R_{i}^{*}$ are distributed around the common mean. Of particular importance for the results of the comparison is the mean square deviation of the participants:(11)$$\sigma^{2}=\overset{n}{\underset{i=1}{\sum }}w_{i}{\left(R_{i}^{*}-1\right)}^{2}=\overset{n}{\underset{i=1}{\sum }}w_{i}\left({R_{i}^{*}}^{2}-2R_{i}^{*}+1\right)=\overset{n}{\underset{i=1}{\sum }}w_{i}{R_{i}^{*}}^{2}-1$$

After replacing the weights by Equation (6), it follows that
(12)$$\sigma^{2}=\frac{\sum_{i=1}^{n}\frac{1}{u_{i}^{2}}\;\sum_{i=1}^{n}\frac{R_{i}^{2}}{u_{i}^{2}}}{{\left(\sum_{i=1}^{n}\frac{R_{i}}{u_{i}^{2}}\right)}^{2}}-1$$

The square root of Equation (12) is considered to be the standard variation interval within which a certain radon activity concentration is measured in the atmospheres of European radon calibration facilities, and is thus a measure of the degree of agreement between the participants. Figure 2 shows the distribution of the single values of $R_{i}^{*}$ around the common mean of 1 and the standard variation interval for each exposure level. The values for the standard and expanded intervals are provided in Table 5. The expanded variation interval is calculated using a coverage factor k=2, which gives a level of confidence of approximately 95%. Although individual participants fell outside the coverage interval, they could not be considered outliers because their deviation was not significant when uncertainties were taken into account.

Already in 2005, a comparison of calibration facilities for radon activity concentration was carried out within the framework of the Euromet Project 657 [3]. The comparison of the expanded variation intervals obtained in this and the previous study are shown in Table 6. Regardless of the different exposure levels, as well as the calculation of the variation intervals, a slight improvement in agreement on the measurements can be assumed. This was particularly evident at higher exposures.

### 4.2. Alternative Determination of Mean Values and Associated Uncertainties

The power-moderated mean method was proposed as an alternative method for calculating the KCRV and the associated standard uncertainty [8]. The method yields results that are generally intermediate between the arithmetic mean and the weighted mean. The power-moderated mean is an efficient and robust estimator of the reference value of a data set and its uncertainty. In particular, the method can be applied to discrepant data sets where the reported uncertainties do not cover the observed dispersion of the data and the condition $\chi_{\mathrm{obs}}^{2}<n-1$ is not satisfied. This must be assumed for the data that were determined for the exposure level of 400 Bq/m^3^.

The calculations of the power-moderated mean were performed using the Excel spreadsheet MET511639suppdata.xlm, which is available for download from the Internet [9]. The automatic algorithm for moderating the relative weighting is used. Table 7 shows the values of the power-moderated mean and the corresponding standard uncertainties calculated with the Excel spreadsheet.

The comparison with the complementary data of the average ratios in Table 4 does not show relevant differences. Only for the exposure level of 400 Bq/m^3^ were slightly larger changes produced for the power-moderated mean method.

The approach provided by the Excel spreadsheet MET511639suppdata.xlm was also used for the identification of extreme values. Extreme values are indicated when the difference between the measured ratio and the power-moderated mean, namely, $d_{i}=R_{i}-R_{w,\mathrm{pm}}$, exceeds the constraint specified by [8]:(13)$$\left|d_{i}\right|>k\cdot u\left(R_{w,\mathrm{pm}}\right)\sqrt{\left(\frac{1}{w_{i}}+1\right)}$$

For the coverage factor k=2, no extreme values were found in the underlying data set. However, if the two participants who traced their measurements back through the factory calibration are included in the data set, some of their results would be classified as extreme values (outliers).

### 4.3. Influence of Climatic Conditions during the Calibrations

Most participants reported their results in conjunction with data on the climatic conditions (temperature, relative humidity and barometric pressure) in the laboratory at the time of exposure. Corrections for standard room conditions (temperature of 20 °C, relative humidity of 50% and air pressure of 1013 hPa) were not required. Only one participant reported his results for standard room conditions.

Exposures at the facilities were conducted in a wide range of climatic conditions that included temperatures (T) from 18 to 28 °C, atmospheric pressures (p) from 950 to 1024 hPa and relative humidities (rH) from below 10% to 63%. Detailed information with data on the climatic conditions is available on the Internet [2]. Figure 3 presents a three-dimensional plot of the climatic conditions that were present for nine of the participants during the exposures. These nine participants provided results for all three exposure levels without correcting them for standard conditions. The different climatic conditions raise the question of what influence they had on the results of this study.

The power of the association between a specified random variable ($R_{i}^{*}$) and a group of independent random variables (T,p,rH) is determined using the multiple correlation method. Details of the procedure and computational results are presented elsewhere [2].

The correlation studies did not reveal a statistically significant indication of a dependence of the comparison value $R_{i}^{*}$ on the climatic conditions during the calibration exercise. However, the pairwise correlation showed a dominant dependence of the ratio $R_{i}^{*}$ on the atmospheric pressure at the exposure level of 6000 Bq/m^3^. This correlation was not expected and is surprising. Since, in most cases, the type of device used by the participants is the same as that of the comparison device, it can be assumed that the climatic conditions affected the devices in the same way and that their effects canceled each other out when calculating the ratios. Therefore, it cannot be excluded that a random correlation was observed. Nevertheless, this finding should be clarified and can only be accepted if it is reproduced by further investigations.

### 4.4. Metrological Traceability and Correlations between the Participants

The participants were requested to provide information on how the metrological traceability of the radon activity concentration was realized. From this information, the chart in Figure 4 was developed, which shows the status of the metrological traceability during the period of the interlaboratory comparison.

The radon activity concentration is a derived quantity that is composed of the base quantities of activity of the gaseous radon and the volume. The volume is the capacity of the enclosed space containing the atmosphere into which radon is released. A secondary reference facility traces both the radon activity and volume back to their respective primary standards and combines the two quantities to form the radon activity concentration.

There were three main branches through which the base quantity of activity was traced back by the participants. The roots of the branches are the national metrology institutes PTB (Brunswick, Germany), LNHB (Paris-Saclay, France) and NIST (Gaithersburg, MA, USA), which hold the primary quantities. PTB (Brunswick, Germany), BfS (Berlin, Germany), IRSN (Paris-Saclay, France) and ENEA (Rome, Italy) act as secondary reference facilities. It should be noted that PTB abandoned its reference chamber in 2016. Facilities, that had used the PTB reference chamber to ensure the metrological traceability will have to undertake a rearrangement after its validity has expired. The BfS reference chamber switched the metrological traceability of the radon activity as of 2020 to LNHB by means of a gas standard. In order to adjust the radon activity concentrations to predetermined values that remain constant over time, emanation sources will become increasingly important in the future.

Figure 4 shows that the secondary reference facilities measuring the derived quantity are not exclusively operated by metrology institutes and thus are not integrated into qualified metrological surveillance. Regular comparisons between the secondary reference facilities should be initiated to ensure the quality of the measurement of the quantity.

## 5. Conclusions

From March 2018 to February 2020, an interlaboratory comparison was conducted in the framework of the EMPIR Project *Metrology for radon monitoring*. In total, 15 calibration facilities from 13 different European countries participated in the interlaboratory comparison. Among those were national metrology institutes and designated institutes, national authorities for radiation protection and participants from universities.

The comparison was conducted by the German Federal Office for Radiation Protection (BfS). An electronic instrument of the type AlphaGUARD was selected as the comparison device, which was provided to the participants in turn. Participants were requested to expose the device to three different levels of radon activity concentration: 400, 1000 and 6000 Bq/m^3^. In certain cases, other exposures were also accepted. The ratio of the value of radon activity concentration established and measured at the participant’s facility to the value that was simultaneously determined with the comparison device in the same atmosphere was used for comparison.

The results showed that, taking the statistical uncertainties into account, the ratios of the radon activity concentrations were identical for all exposure levels and for the summary of all levels including singular exposures. The radon activity concentrations that were determined by European calibration facilities that complied with metrological traceability requirements were consistent with each other and had common mean values. The deviations from these values were normally distributed. It can be assumed that the radon activity concentration that was measured by the European calibration facilities fluctuated around a collective mean value. Its interval of variation was a measure of the degree of agreement between the participants. For exposures above 1000 Bq/m^3^, the variation was about 4% for a level of confidence of approximately 95% (k=2). For lower exposure levels, the variation increased to about 6% at 400 Bq/m^3^.

The participants performed their measurements under different climatic conditions. Correlation studies revealed no statistically significant indication of a dependence of the comparison value on the climatic conditions. However, a correlation was found between the comparison value and the atmospheric pressure for the exposure level of 6000 Bq/m^3^. This finding could not be clarified in the present study and requires further investigation.

The European radon calibration facilities traced their measurements back to primary quantities that were maintained at the national metrology institutes PTB (Brunswick, Germany), LNHB (Paris-Saclay, France) and NIST (Gaithersburg, MD, USA). The results of the interlaboratory comparison show that metrological traceability and calibration of instruments must be carried out according to uniform and generally recognized standards. Where these standards are not met, sufficient confidence in the measurement results cannot be established.

The interlaboratory comparison of European radon calibration facilities is a powerful tool to detect discrepancies in metrological traceability and to ensure the quality of radon measurements. It is strongly recommended to carry out interlaboratory comparisons regularly.

## Figures and Tables

**Figure 1 ijerph-18-12150-f001:**
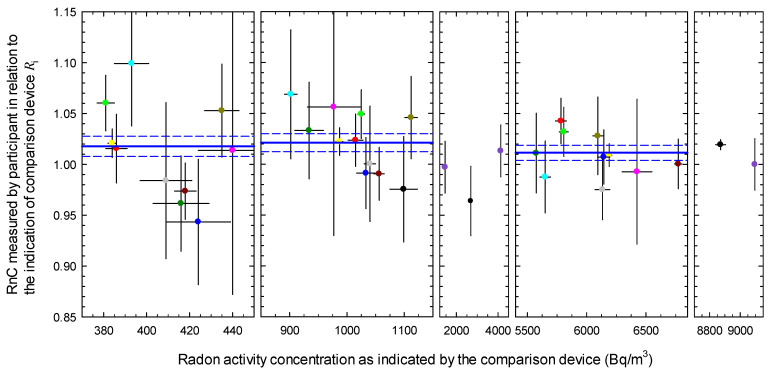
Ratio $R_{i}$ of the mean radon activity concentration (RnC) that was determined by each participant to that of the comparison device given for the different exposures; error bars indicate the standard uncertainties of the reported values, results of the same participant are indicated by the same color, blue straight lines indicate the uncertainty-weighted average ratio $R_{w}$ and dashed blue lines cover the range of the standard uncertainty.

**Figure 2 ijerph-18-12150-f002:**
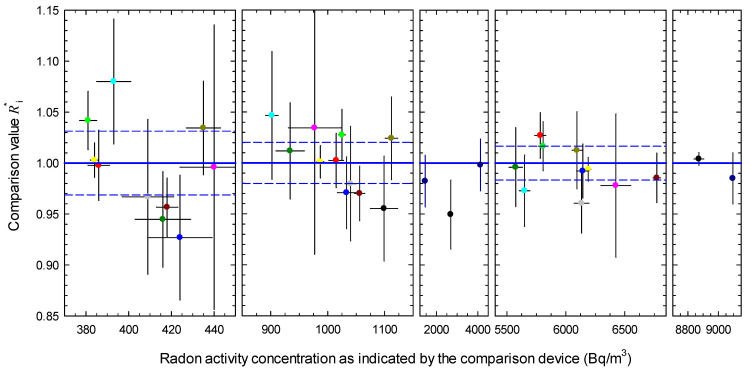
Comparison values given as modified ratios ($R_{i}^{*}$) and attributed uncertainties; blue straight lines represent a common mean of 1, dashed blue lines cover the standard variation interval (coverage factor k=1) according to Table 5, error bars indicate the standard uncertainties of the reported values, the results of the same participant are indicated by the same color and the color assigned to the respective participant is the same as in Figure 1.

**Figure 3 ijerph-18-12150-f003:**
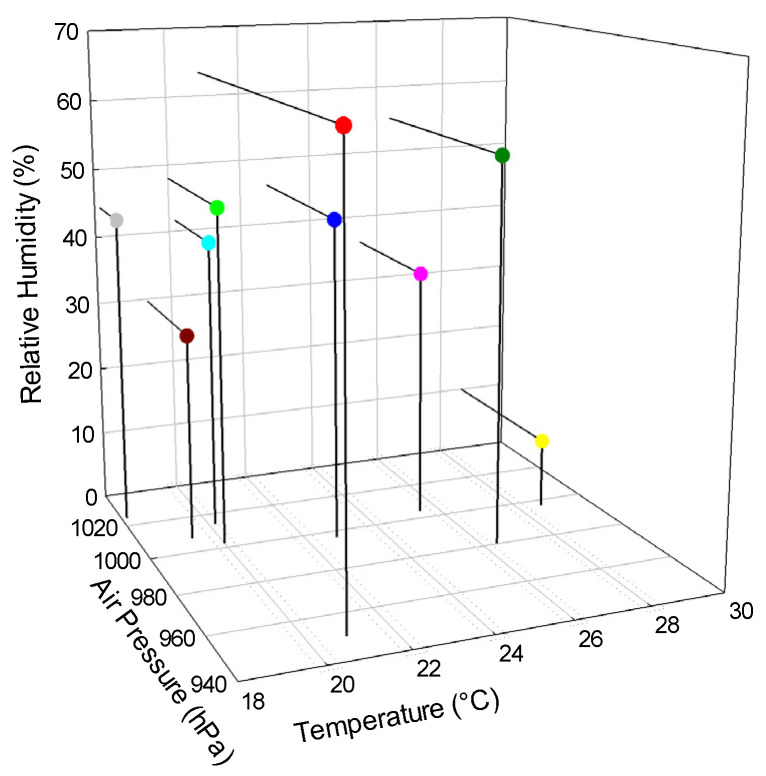
Values of temperature, air pressure and relative humidity during the exposures, as reported by the participants; the colors used for the participants are the same as in Figure 1 and Figure 2.

**Figure 4 ijerph-18-12150-f004:**
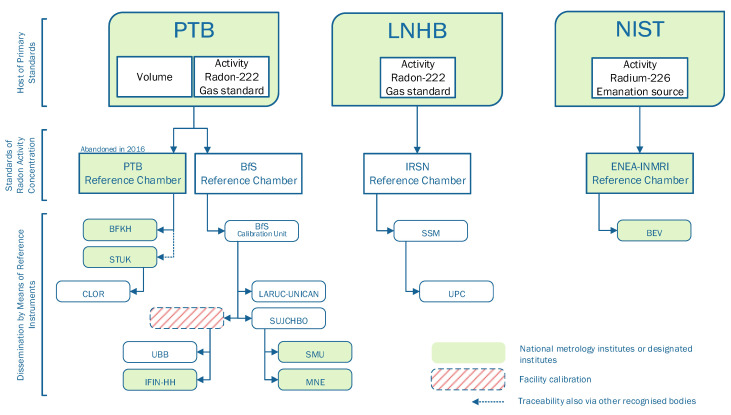
Chart of metrological traceability of European calibration facilities for radon: status at the start of the interlaboratory comparison (2018).

**Table 1 ijerph-18-12150-t001:** Calibration facilities participating in the interlaboratory comparison (sorted alphabetically by country).

Short Name	Affiliation	Country
BEV-PTP	Physikalisch-technischer Prüfdienst, Bundesamt für Eich- und Vermessungswesen Arltgasse 35, 1160 Wien	Austria
SUJCHBO	Státní ústav jaderné, chemické a biologické ochrany Kamenna 71, 262 31 Milin	Czech Republic
STUK	Radiation and Nuclear Safety Authority Laippatie 4, 00880 Helsinki	Finland
IRSN	Institut de Radioprotection et de Sûreté Nucléaire 31 avenue de la Division Leclerc, 92262 Fontenay-aux-Roses	France
BfS (Coordinator)	German Federal Office for Radiation Protection Köpenicker Allee 120–130, 10318 Berlin	Germany
BFKH	Budapest Főváros Kormányhivatala Németvölgyi út 37-39, 1024 Budapest	Hungary
ENEA	ENEA-INMRI, via Anguillarese, 301 - 00123 Roma	Italy
MNE	Bureau of Metrology Arsenija Boljevića bb, 81000 Podgorica	Montenegro
CLOR	Central Laboratory for Radiological Protection Konwaliowa 7, 03-194 Warsaw	Poland
IFIN-HH	Institutul National de Cercetare-Dezvoltare pentru Fizica si Inginerie Nucleara “Horia Hulubei” 30 Reactorului St., 077125 Magurele, Ilfov County, POB MG-6	Romania
UBB	“CONSTANTIN COSMA” Radon Laboratory, Babes—Bolyai University, Faculty of Environmental Science and Engineering Fantanele 30, 400294 Cluj-Napoca	Romania
SMU	Slovak Institute of Metrology, Department of Ionizing Radiation Karloveská 63, 842 55 Bratislava	Slovak Republic
LaRUC	Radon Group, Laboratory of Environmental Radioactivity of the University of Cantabria (LaRUC) C/Cardenal Herrera Oria S/N, 39011 Santander, Cantabria	Spain
UPC	Laboratory of ^222^Rn studies (LER) of the Institut de Tècniques Energètiques (INTE) of the Universitat Politècnica de Catalunya (UPC), Campus Diagonal Sud, Edificio PC (Pavelló C) Av. Diagonal, 647, 08028 Barcelona	Spain
SSM	Swedish Radiation Safety Authority Solna strandväg 96, 171 16 Stockholm	Sweden

**Table 2 ijerph-18-12150-t002:** Nominal levels of the radon activity concentrations for the exposure of the comparison device.

No.	Nominal Value (Bq/m^3^)	Range of Accepted Deviation (Bq/m^3^)
1	400	350–450
2	1000	900–1100
3	6000	5500–6500

**Table 3 ijerph-18-12150-t003:** Chi-squared consistency check for the different radon levels and for all levels.

Exposure Level	No. of Measurements *n*	$\chi_{\mathrm{obs}}^{2}$ (Observed)	$\chi_{0.05;n-1}^{2}$ (Tabulated)
400 Bq/m^3^	10	10.45	16.92
1000 Bq/m^3^	11	5.49	18.31
6000 Bq/m^3^	10	5.16	16.92
All levels including singular exposures	36	25.17	49.80

**Table 4 ijerph-18-12150-t004:** Uncertainty-weighted average ratio and its standard uncertainty for the different exposure levels.

Exposure Level	Uncertainty-Weighted Average Ratio $R_{w}$	Standard Uncertainty Associated with $R_{w}$ $u\left(R_{w}\right)$
400 Bq/m^3^	1.018 ^1^	0.010
1000 Bq/m^3^	1.021 ^1^	0.009
6000 Bq/m^3^	1.012 ^1^	0.007
6000 Bq/m^3^ including singular exposures	1.015	0.004
All levels including singular exposures	1.016	0.003

^1^ Indicated by the blue straight lines in Figure 1.

**Table 5 ijerph-18-12150-t005:** Standard and expanded variation intervals for the measurement of the radon activity concentration at different exposure levels.

Exposure Level	Standard Variation Interval (%)	Expanded Variation Interval (%)
400 Bq/m^3^	3.2 ^1^	6.3
1000 Bq/m^3^	2.0 ^1^	4.0
6000 Bq/m^3^	1.7 ^1^	3.4
6000 Bq/m^3^ including singular exposures	1.2	2.4
All levels including singular exposures	1.7	3.4

^1^ Indicated by blue dashed lines in Figure 2.

**Table 6 ijerph-18-12150-t006:** Expanded variation intervals for the measurements of the radon activity concentration obtained in this study and the EUROMET Project 657.

Exposure Level	This Study (All Participants)	EUROMET Project 657 (Final Report 2005)
400 Bq/m^3^	0.063	--
1000 Bq/m^3^	0.040	0.057
3000 Bq/m^3^	--	0.075
6000 Bq/m^3^	0.034	--
10,000 Bq/m^3^	--	0.081

**Table 7 ijerph-18-12150-t007:** Power-moderated mean and its standard uncertainty for the different exposure levels, which were calculated using the Excel spreadsheet MET511639suppdata.xlm.

Exposure Level	Power-Moderated Mean $R_{w,\mathrm{pm}}$	Standard Uncertainty Associated with $R_{w,\mathrm{pm}}$ $u\left(R_{w,\mathrm{pm}}\right)$
400 Bq/m^3^	1.016	0.013
1000 Bq/m^3^	1.021	0.009
6000 Bq/m^3^	1.011	0.008
6000 Bq/m^3^ including singular exposures	1.014	0.004
All levels including singular exposures	1.016	0.004

## Data Availability

The data set, as well as further analyses, are available online at http://metroradon.eu/wp-content/uploads/2017/06/16ENV10-MetroRADON-D7-final_accepted.pdf. (accessed on 21 July 2021).

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
