# Peer review of "The Metrological Traceability, Performance and Precision of European Radon Calibration Facilities"

_ijerph, 2021, doi:10.3390/ijerph182212150_

Round 1

Reviewer 1 Report

This paper is intercalibration procedure with many participants. There two significant issues that have been to be addressed prior to accept the substantial intecalibration results that they present with interesting ways.

I suggest that they solve these issues and I will be very glad to review it again.

At this phase I suggest major revision.

Below the authors may find some initial comments.

Abstract: 

1.measurand —> measured. Please change it everywhere

2.The phrase “The results of the comparison …a collective mean value. “ is obvious for stochastic procedures. Please withdraw it or reword.

Keywords:

3.There are two “;” of green background

Introduction

4.”calibration bodies” —> “Intercalibration participants”

5.”in EU member States” —> “by the participating Institutes”. EU member states is quite general and needs many more procedures. Please reword as indicated.

6.Significant:”validate” —> compare. Validation and verification are concepts strictly referring to model building and simulation checking. Please reword. Please avoid using it further. The term inter comparison or comparison refers to what is actually implemented.  Also in title

7.”preferably represent”. The Institutes either are the National 

8.Significant: The main problem in the calibration check of Alpha Guard Pro (gold standard as the authors admit) is the implementation of constant exposures. Since the authors also refer to it as a “measurement uncertainty” it is crucial that they present data regarding this issue. In any case, the should discuss it in detail and estimate the errors in exposure.  Obviously the constant calibration factor is a vague concept.

9.Significant: Since fully aware of AG, the instrument provides QA values. Since the value (4+/-5) Bq/m3 is not acceptable (minus concentration does not exist, this range might not be stochastic at all!), the authors should provide adequate information. In this sense the footnote 1 should b deleted.

Reviewer 2 Report

This work summarizes the methodology and main results obtained in a European radon calibration facilities interlaboratory comparison conducted between 2018 and 2020 included in the European project Metrology for radon monitoring (MetroRADON). 15 different calibration facilities took part in this exercise to validate their metrological traceability as well as their performance and precision calibrating radon measuring instruments. The instruments were tested in the range of (300 - 10000) Bq/m^3 in three different windows representing low, medium and high activity concentration. Given that it was impossible to assure that all laboratories were using the same reference value for radon activity concentration, a measuring instrument provided by the exercise coordinator was provided as a comparison device, used for all participant in all their measurements along their instruments.

This is a clearly and well written manuscript where both the results and procedure used to obtain them will be of interest. All the relevant information to properly describe the followed protocol is covered in the manuscript.  The relevant figure of merit in this study was the ratio between the lectures of tested and comparison instrument as well as their corresponding uncertainty.  Several statistical analysis approaches, all properly explained, were used to obtain the average ratio which should be used as reference value of the exercise. In all cases the results were satisfactory, which could not directly trace their instruments to providing evidence of the competence as calibrating services of some of the main European radon calibrating facilities. 

I only have two minor suggestions:

1. In Figure 1 there is no reference provided to indicate the value of the uncertainty for the weighted average R_w. Although it is already explained in lines 310-315 that this uncertainty is too small and not appropriate to measure the degree of agreement between participants, it might be useful to include it in Figure 1 to compare with Figure 2. 

2. I'm not sure about how easy it is to interpret and properly visualize the results included in Figure 3. This 3D plots are sometimes difficult to interpret as it strongly depends on the chose point of view. The same information could be included using a table, with additional columns to report the corresponding R values. In case that the authors choose to keep this figure, I would advise to use the same color code that in figures 1 and 2 so it would at least be possible to connect all results.

In summary, I think that this is an interesting work which clearly deserves publication and for this reason I recommend its publication in the International Journal of Environmental Research and Public Health.

Reviewer 3 Report

Dear Authors

Check the following technical and language related comments: 

Technical comments:

  1. Why only 3 different levels of 400 Bq m-3, 1000 Bq m-3 and 6000 Bq m-3 values have been selected to expose the device?
  2. Why this performance and precision experiments are not performed below 100 Bq m-3 or 200 Bq m-3, as per the recommended values given by ICRP and UNSCEAR? In many countries the value of radon concentration is lower than 100 - 200 Bq m-3.
  3. It is suggested to be mentioned all radon sources and arrangement of the generation of radon gas in different radon calibration facility.
  4. In the Figure 1 and Figure 2, why only three values are showing in between 2000 Bq m-3 to 4000 Bq m-3 as compare to their neighbor window?
  5. In the section 4.2, line number: 352-358, Do you have checked the normality of your data. In most of the cases of radionuclides and radon measurements, the outcome results show non-normal distribution around their mean (the link of datasheets are not working). It is better to include Q-Q plots and discuss the skewness and kurtosis to know the exact distribution of data.
  6. The radon decay products as Po-214, Po-212 are actually dose given to respiratory tract. Why you have not included their decay measurements in these types of performance and precision experiments.

Language related minor comments

  1. In the throughout manuscript, write the radon in single way in abstract as well as in paper, for exp: either of 222Rn or simple radon. Write its full name with formula when it first time appear in the manuscript.
  2. There are some written language errors in the manuscript. For exp: in line number 125, Bq/m3(1 . Try to avoid all these types of errors.
  3. Many same lines have been found in abstract and discussion and conclusion section (For exp: 39 – 41, 450 - 453).

Round 2

Reviewer 1 Report

The paper has been substantially enhanced in terms of expressions and declarations of the methodologies. The authors reply and rebuttal is sufficient. The paper adds new data to the EU radon problem and is significant in terms of laboratory participation and analysis.

It is suggested for publication.